# M+: Extending MemoryLLM with Scalable Long-Term Memory

Yu Wang [1]   Dmitry Krotov [2,3]   Yuanzhe Hu [1]   Yifan Gao [4]   Wangchunshu Zhou [5]   Julian McAuley [1]
Dan Gutfreund [2,3]   Rogerio Feris [2,3]   Zexue He [1,2,3]

## Abstract

Equipping large language models (LLMs) with latent-space memory has attracted increasing attention as they can extend the context window of existing language models. However, retaining information from the distant past remains a challenge. For example, MemoryLLM (Wang et al., 2024a), as a representative work with latent-space memory, compresses past information into hidden states across all layers, forming a memory pool of 1B parameters. While effective for sequence lengths up to 16k tokens, it struggles to retain knowledge beyond 20k tokens. In this work, we address this limitation by introducing M+, a memory-augmented model based on MemoryLLM that significantly enhances long-term information retention. M+ integrates a long-term memory mechanism with a co-trained retriever, dynamically retrieving relevant information during text generation. We evaluate M+ on diverse benchmarks, including long-context understanding and knowledge retention tasks. Experimental results show that M+ significantly outperforms MemoryLLM and recent strong baselines, extending knowledge retention from under 20k to over 160k tokens with similar GPU memory overhead. We open-source our code at https://github.com/wangyu-ustc/MemoryLLM.

## 1. Introduction

The integration of memory modules into large language models (LLMs) has gained increasing attention (Wang et al., 2024b). Existing approaches for constructing memory modules can be broadly divided into two main categories: (1) Token-level memory (Packer et al., 2023; Modarressi et al., 2024), where memory is represented as structured text, en-

abling direct retrieval and manipulation of information at the token level; and (2) Latent-space memory, where memory is stored as high-dimensional vectors in the hidden space, offering a more abstract and compact representation of information. Token-level memory provides adaptability (the base model can be easily replaced) and interpretability (text-based format is easy to understand for humans). However, such text-based memory could be redundant as text format may not be the most compressed method for representing information (Bellard, 2021; Belcak & Wattenhofer, 2024; Rahman et al., 2024), and resolving conflicting information in text-based memory can be challenging (Pham et al., 2024). Meanwhile, as noted by Fedorenko et al. (2024); Hao et al. (2024), human reasoning often transcends the token level, leveraging deeper, integrated representations akin to latent spaces.

In contrast, Latent-Space Memory offers unique advantages: (1) **Efficient Compression**: Information is compressed into hidden states (Wang et al., 2024a), internalized into model parameters (Wang et al., 2024c), or stored in a more compact latent space (Das et al., 2024). These methods reduce storage overhead, with some approaches even embedding knowledge directly into model parameters, eliminating the need for external storage (Wang et al., 2024c). (2) **End-to-End Training**: Latent-space memory can be involved in gradient-based optimization, allowing it to be updated and refined during training. This enables the integration of memory into the training loop (Yin et al., 2024; Wang et al., 2023; Ge et al., 2024). (3) **Similarity to Human Memory**: As suggested by Fedorenko et al. (2024) and Hao et al. (2024), human reasoning relies on integrated representations beyond discrete tokens, akin to latent spaces. By encoding knowledge in latent representations, the methods with latent-space memory can more closely mimic the mechanisms of human memory, which store information within neural activations.

In this paper, we focus on Latent-Space Memory. MemoryLLM (Wang et al., 2024a), as a representative work in this category, enhances a transformer-based language model by incorporating a large number of memory tokens into each layer, creating a memory pool with 1 billion parameters. This framework introduces a carefully designed update and generate process, achieving superior performance compared

---

Work done during the internship at MIT-IBM Waston Lab.
[1]UC, San Diego [2]MIT-IBM Waston Lab [3]IBM Research [4]Amazon [5]OPPO. Correspondence to: Yu Wang <yuw164@ucsd.edu>, Zexue He <Zexue.He@ibm.com>.

to the backbone model Llama-2-7B and other long-context methods. However, MemoryLLM faces limitations in recalling information injected beyond 20k tokens, restricting its long-term retention capabilities. To address this limitation, we propose **M+**, a novel model incorporating a long-term memory mechanism alongside MemoryLLM. Unlike previous approaches such as H2O (Zhang et al., 2023) and SnapKV (Li et al., 2024), which store keys and values from past contexts and perform retrieval separately for each query head and layer—leading to high latency—M+ optimizes retrieval in the space of hidden states via co-training the retriever and the language model. This allows M+ to retrieve only once per layer for all query heads, significantly improving efficiency. Furthermore, as the long-term memory is stored on the CPU, M+ significantly extends long-term retention capabilities without increasing GPU memory usage.

We evaluate M+ across a diverse set of benchmarks, including tasks such as long-book understanding, knowledge retention, and question answering on relatively short documents. Experimental results demonstrate that M+ achieves significant performance improvements in all long benchmarks compared to previous memory-based methods while operating within the same or even smaller inference memory budget. In summary, our contributions are as follows:

- We enhance MemoryLLM by incorporating a long-term memory mechanism and introducing a co-trained retriever for efficient and effective memory retrieval.

- We design a specialized data curriculum for long-context training, enhancing the long-context modeling ability of M+.

- Through extensive experiments on multiple benchmarks, we demonstrate that M+ significantly outperforms the baselines while maintaining a similar or reduced GPU memory footprint.

## 2. Related Work

We classify memory-based methods into two categories: Token-Level Memory and Latent-Space Memory, which is similar to the categorizations in Yin et al. (2024) where they classify methods into implicit memory and explicit memory.

### 2.1. Token-level Memory

Token-level memory refers to memory structures represented in textual forms, which can include raw context, summaries (Zhong et al., 2023; Zhou et al., 2023), knowledge graphs (Packer et al., 2023; Gutiérrez et al., 2024), organized text with hierarchical or graph structures (Packer et al., 2023; Chen et al., 2024), or databases (Hu et al., 2023). Methods such as MemoryBank (Zhong et al., 2023), RecurrentGPT (Zhou et al., 2023) incorporate multiple components of memory, including both raw conversational data and summaries. MemGPT (Packer et al., 2023) pro-

poses treating context and memory as analogous to traditional memory in operating systems, enabling more flexible and organized memory structures. These approaches typically rely on text embeddings for memory retrieval, where queries can originate from either the current conversation turn (Zhong et al., 2023; Zhou et al., 2023) or queries generated by the language model itself (Packer et al., 2023). In contrast, ChatDB (Hu et al., 2023) stores knowledge in a database and performs retrieval using SQL queries, while MemLLM(Modarressi et al., 2024) fine-tunes the model to generate function calls that initiate searches within a knowledge graph, referred to as "Triple Memory" by Modarressi et al. (2024). These methods generally offer benefits such as modularity (with the exception of MemLLM, which requires fine-tuning) and interpretability (Yin et al., 2024), allowing for potential integration with external systems (Wu et al., 2022a). However, these approaches have limitations. Some require storing the raw text, which is not the most compressed method to store information (Rahman et al., 2024; Bellard, 2021; Belcak & Wattenhofer, 2024). Others store knowledge in the form of triplets, which may be unsuitable for representing complex conversations that are difficult to convert into triplets (Wang et al., 2024d).

### 2.2. Latent-Space Memory

Latent-space memory stores information in a compressed format, embedding knowledge into soft prompts (Rakotonirina & Baroni, 2024), hidden states (Khandelwal et al., 2019; Bulatov et al., 2022; 2023; Wang et al., 2024a), model parameters (Wang et al., 2024c), or an external latent space (Das et al., 2024), among other methods. Some approaches use memory slots to encode information (Al Adel & Burtsev, 2021), while others rely on key-value caches stored in memory pools for future retrieval (Wu et al., 2022b; Wang et al., 2023; He et al., 2024; Park & Bak, 2024). Notably, CamelLoT (He et al., 2024) and Memoria (Park & Bak, 2024) incorporate forgetting mechanisms to better emulate human memory. Similarly, MemoryLLM (Wang et al., 2024a) compresses knowledge into hidden states and employs random dropping to prevent unbounded memory growth. The $M^3$ method (Yang et al., 2024) also stores memory in the hidden-state space, archiving a vast pretraining dataset comprising $1.1 \times 10^8$ text chunks. Distinct from methods that utilize hidden states or key-value caches, Larimar (Das et al., 2024) introduces a memory matrix that supports read and write operations, demonstrating effectiveness in knowledge-editing tasks. Furthermore, SELF-PARAM (Wang et al., 2024c) explores embedding knowledge directly into model parameters without degrading the model's capabilities or requiring additional parameters. These latent-space memory techniques have shown promising results across various downstream tasks. By saving information in a compressed format and leveraging retrieval during generation, they enable substantial expan-

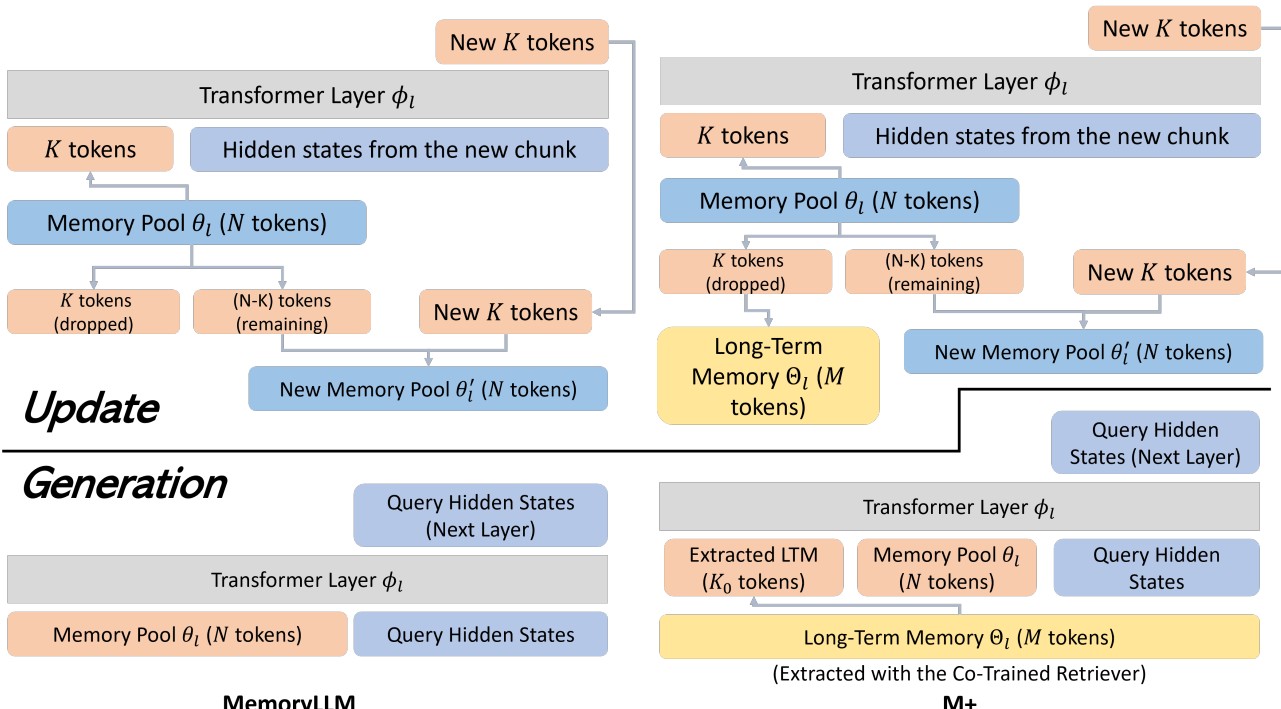

*Figure 1.* The left side shows the Update and Generation Process of MemoryLLM (Wang et al., 2024a). We process the chunk with $\phi_l$ to obtain new $K$ tokens during the update process, which is perceived by $\phi$ using cross-attention during the generation process. The right side shows the Update and Generation Process of M+. For layer $l$, during Update, the old memory pool $\theta_l$ is split into two parts: $K$ dropped tokens and $N - K$ remaining tokens. The dropped tokens are stored in the long-term memory $\Theta_l$ while the remaining tokens and new $K$ tokens are combined to obtain the new memory pool $\theta'_l$. Then during generation, we use our co-trained retriever to retrieve tokens from $\Theta_l$, which is fed into the transformer layer $\phi_l$ along with the short-term memory $\theta_l$ and the query hidden states. The major difference between MemoryLLM and M+ is the introduction of Long-Term Memory $\Theta_l$.

sions of the context window without incurring excessive GPU memory costs. Despite the advantages and potential of Latent-Space Memory, existing methods within this category typically fall short when dealing with extremely long input (Das et al., 2024; Wang et al., 2024c;a; He et al., 2024). In contrast, M+ can have much longer retention compared to existing methods.

## 3. Methodology

### 3.1. Preliminaries

We first introduce the structure of MemoryLLM (Wang et al., 2024a), which serves as the base structure of M+. MemoryLLM comprises two main components: $\theta$ (the memory pool) and $\phi$ (a transformer-based decoder-only language model). The memory pool $\theta$ consists of $L$ layers: $\{\theta_l\}_{l=1}^L$ where $L$ is the number of layers in the transformer $\phi$. For every layer, $\theta_l$ has $N$ memory tokens, where each token is a vector in $\mathbb{R}^d$, with $d$ representing the hidden size of the language model. During the update process, the last $K$ tokens from the $l$-th layer's memory pool, $\theta_l$, are extracted and combined with the chunk to be injected. The resulting new $K$ tokens are then merged back into $\theta_l$ (illustrated in

Figure 1). Merging is achieved by randomly dropping $K$ tokens from $\theta_l$ and appending the new $K$ tokens to the end. During generation, the memory pool $\theta_l$ is perceived using cross-attention.

### 3.2. Equipping MemoryLLM with Long-Term Memory

In this section, we explain how we instantiate the long-term memory and how it integrates with the language model $\phi$ and the original memory pool $\theta$ in MemoryLLM. In this paper, we term the original memory pool $\theta$ as short-term memory to distinguish it from the new long-term memory.

#### 3.2.1. MEMORY STRUCTURES

We denote the long-term memory as $\Theta$. Similarly, it has $L$ layers $\{\Theta_l\}_{l=1}^N$. Each layer has a long-term memory pool where the size is flexible. We specify a maximum size for the long-term memory. The maximum size of the long-term memory is denoted as $M$ and the size of long-term memory is flexible. In practice, we choose $M$ to be 150k. Then we introduce the update and generate process of M+:

**Update Process** During the update process, note that in the original MemoryLLM, $K$ tokens are dropped from $\theta$ during updates and are permanently discarded. In M+, the

dropped $K$ tokens are instead stored in the long-term memory $\Theta$, ensuring their retention for extended durations (as illustrated in Figure 1). We assign each token the variable "age" so that after retrieving tokens from $\Theta$ we can sort these tokens according to age, ensuring that the tokens are chronologically ordered. As for the new $K$ tokens, they are obtained with the same process as in MemoryLLM, described in Figure 1. When the memory tokens in the long-term memory reach the maximum capacity, i.e., $M$ tokens, we would drop the tokens with the largest ages.

**Generation Process** During generation, at each layer, we extract $K_0$ tokens from the long-term memory $\Theta_l$ using a retrieval mechanism described below, sort them by their ages, and concatenate them with the short-term memory $\theta_l$. This allows the query hidden states to access both the extracted long-term memory and the short-term memory using cross-attention, enabling the query to retrieve relevant information from the memory.

**Multi-LoRA Design** In our training, we use two sets of LoRA weights, one is activated during the update process, and the other is activated during the generation process (as shown in Figure 1). Intuitively, the update process compresses the information (similar to writing) while the generating process loads the information (similar to reading), thus having two LoRA weights could potentially make learning easier for our model. This is similar to the intuition in T5 where they find sharing the weights of encoder and decoder leads to slightly inferior performances (See Table 2 in Raffel et al. (2020)).

### 3.2.2. RETRIEVER DESIGN AND TRAINING

**Retriever Design** The retriever has two projectors: query projector $f_q$ and key projector $f_k$, which are all instantiated with a two-layer perceptron. The output dimension of both projectors, denoted as $d_{proj}$, is set to be a small number. In our experiments, we set $d_{proj}$ to be $d/20$ where $d$ is the hidden size of the language model $\phi$. When dropping tokens from $\theta_l$ into $\Theta_l$ (as shown in Figure 1), we apply $f_k$ on top of the dropped memory tokens, thus we need an additional pool storing all the key vectors corresponding to the memory tokens in $\Theta_l$. Note that the key vectors are the output from $f_k$, and are of dimension $d_{proj}$, requiring little additional memory footprint compared to the long-term memory. During generation, given the hidden states from the query, we apply $f_q$ on the hidden states to get query vectors and use them to retrieve tokens from $\Theta_l$ according to the dot product between query vectors and key vectors.

**Training the Retriever** To train the retriever, we first split a document $x$ into $n$ chunks $x_1, x_2, \cdots, x_n$ and we inject $\{x_1, \cdots, x_{n-1}\}$ into the short-term memory. Then we can track the embeddings in the short-term memory that are related to $x_1, \cdots, x_{n-1}$ which we denote as $\theta_+$. Then the memory tokens that are not related to $x_1, \cdots, x_{n-1}$, i.e., the tokens that were there before injecting $x_1, \cdots, x_{n-1}$,

are denoted as $\theta_-$. After that, we run a forward pass on $x_n$ to obtain the hidden states $h_n$ (Note that this is a general notation for all layers). For the hidden states $h_n$ in each layer, we train the retriever using the following objective:

$$\min_{f_q, f_k} -\log(p_+) - \log(1 - p_-),$$
$$\text{where } p_+ = \langle f_q(h_n), f_k(\theta_+) \rangle,$$
$$p_- = \langle f_q(h_n), f_k(\theta_-) \rangle,$$

i.e., we are maximizing the distance between $h_n$ and $\theta_-$ while minimizing the distance between $x_n$ and $\theta_+$ after applying $f_q$ and $f_k$ on $h_n$ and $\theta$ respectively.

### 3.2.3. TRAINING DETAILS

**Setting Configurations** We build M+ on top of Llama-3.1-8B (Dubey et al., 2024) and train it using eight A100 GPUs. We tried `FSDP`, `deepspeed-stage-2`, and `deepspeed-stage-3` and we finally choose `deepspeed-stage-2` due to resource limitation and library incompatibility (see the details in Appendix A). Specifically, we set $K = 256$, $N = 10240$ ($N$ is the number of tokens in the short-term memory, see Section 3.1), and the number of tokens of extracted LTM in Figure 1 is set to 2,560. The generation window (i.e., the maximum length of generation) is set to be 2,048. Thus maximally we have the attention matrix of shape $(12, 800+2, 048)$ by $2, 048$, which is fit into eight A100 GPUs using `deepspeed-stage-2`. Although Llama-3.1-8B can practically handle a 128k context window, it went through much more extensive training that we cannot afford. Should we have more GPUs and more budget, M+ could also be scaled to 128k level. Given the constraint of GPU resources, we have scaled to 12,800 memory tokens and 2,048 generation context window, relying on Llama-3.1-8B's capability on a context window of $12, 800 + 2, 048 = 14, 848$ tokens. Thus, for fair comparisons, we mainly focus on Llama-3.1-8B-**16k** as the baseline.

### 3.2.4. DATA CURRICULUM

The training process consists of three stages:

**Continual Training of MemoryLLM (Stage 1)** Different from (Wang et al., 2024a) which starts from the backbone model Llama-2-7B, we start with the backbone model Llama-3.1-8B, which serves as $\phi$ as shown in Figure 1. We equip $\phi$ with $N = 12, 800$ memory tokens in each layer and set the generation context window as 2,048. We first continually train $\phi$ equipped with $\theta$ on the dataset `fineweb-edu` (Penedo et al., 2024) for 1,200,000 steps over four weeks, establishing a strong foundation for handling short documents. This training stage consists of three key sub-tasks as outlined in MemoryLLM (Wang et al., 2024a) (see details in Appendix D).

**Long-Context Modeling with Long Documents (Stage 2)** Since most of the `fineweb-edu` dataset (used in Stage

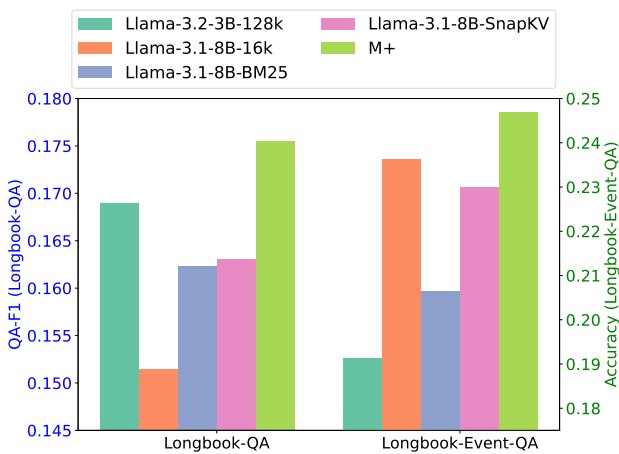

*Figure 2.* Overall Performance Comparison Longbook Question Answering. Best viewed in colors.

1 training) are short documents under 4k tokens, we need to train on longer documents to enhance the model's long-context modeling abilities. Thus, we extract documents from SlimPajama that range from 4k to 64k tokens and split them into four categories based on their lengths: `4k-8k`, `8k-16k`, `16k-32k`, `32k-64k`. The statistics of obtained dataset is shown in Appendix C. For each length range, we randomly sample 200,000 examples, and they are combined with a snapshot of `fineweb` in equal proportions (1:1:1:1:1), with each subset contributing to 20% of the total data. We set this proportion to upsample longer documents, which is important for long context modeling as suggested by Fu et al. (2024). Training runs for one epoch with around one week using the same training tasks in Stage 1.

**Training with long-term memory (Stage 3)**  Building on Stage 2, we introduce long-term memory to enhance M+. Note that in Stage 1 and Stage 2, there is only the short-term memory $\theta$ where each layer $\theta_l$ has 12,800 tokens. In stage 3, we adjust the configuration by setting $\theta_l$ to 10,240 tokens and retrieving $K_0 = 2,560$ tokens from the long-term memory, maintaining a total of 12,800 memory tokens as in the previous stages. Now the structure of the memory tokens becomes slightly different, as 2,560 tokens are from the long-term memory, we design Stage 3 to ensure the model $\phi$ understand the tokens from long-term memory – we continuously train from the checkpoint obtained after Stage 2 on a newly constructed dataset sampled from the same long documents extracted from SlimPajama but distinct from the instances used in Stage 2.

## 4. Experiments

### 4.1. Long Book QA and Event QA

#### 4.1.1. EXPERIMENTAL SETTINGS

We evaluate our model on two datasets designed to test long-context understanding and long-term memory capabilities:

**LongBook-QA**: This dataset is part of ∞-Bench (Zhang et al., 2024) and consists of 351 tuples in the format (`book`, `question`, `answer`). Each book has an average input length of 192k tokens. The task requires answering questions based on the entire book, and we use the QA-F1 score as the evaluation metric.

**LongBook Event QA**: We propose this new benchmark to evaluate the model's ability to recall past events and reason chronologically. This dataset is constructed as follows: (1) We use the Named Entity Recognition (NER) tool from `SpaCy` to identify the ten most frequently mentioned characters in each of the first five books from the LongBook-QA dataset. (2) Each book is divided into 4096-token chunks in chronological order, and events experienced by the main characters are extracted using `gpt-4o`. This results in event lists with 1,016, 221, 644, 348, and 409 events for the five books, respectively. (3) For each event, we construct a multi-choice question-answering task by prompting `gpt-4o` to generate five fake events as distractors. The model is provided with the book text, five past events, and asked to identify the ground-truth event from six options. We use accuracy as the evaluation metric.

We compare M+ against the following baselines: (1) **Llama-3.1-8B-16k**: The original model with context window fixed as 16k. (2) **Llama-3.1-8B-SnapKV**, We processes a 32k token input and dynamically selects 16k key-value caches from the saved 32k caches with the techniques introduced from SnapKV (Li et al., 2024). SnapKV incurs significant memory overhead, as illustrated in Section 4.2. (3) **Llama-3.1-3B-128k**: A 3B parameter version of the Llama3 series. This model uses a 128k context window and consumes comparable GPU memory to M+ because of its smaller size.

(1) **Llama-3.1-3B-128k**: A 3B parameter version of the Llama3 series. This model uses a 128k context window and consumes comparable GPU memory to M+ because of its smaller size. (2) **Llama-3.1-8B-16k**: The original model with context window fixed as 16k. (3) **Llama-3.1-8B-BM25**: We use BM25 as the retriever. Specifically, we process the entire book by dividing it into chunks of 4,096 tokens and retrieve four relevant chunks for each question. The model Llama-3.1-8B is required to answer the question with four retrieved chunks. (4) **Llama-3.1-8B-SnapKV**, We processes a 32k token input and dynamically selects 16k key-value caches from the saved 32k caches with the techniques introduced in SnapKV (Li et al., 2024). SnapKV incurs significant memory overhead, as illustrated in Section 4.2.

We present the primary results in the main paper, while deferring extended discussions and supplementary experiments to the appendix. These include: similarities to attention-based retrieval methods (Appendix E.1); the structure of long-term memory (Appendix E.2); latency and memory consumption when scaling (Appendix E.3); FLOPs comparison (Appendix E.4); and the interpretability of memory

vectors (Appendix E.5).

### 4.1.2. EXPERIMENTAL RESULTS

The results for both benchmarks are shown in Figure 2. From the results, we observe the following: (1) M+ consistently outperforms all baselines, demonstrating its efficiency and effectiveness. In LongBook-QA, M+ achieves superior QA-F1 scores even while processing the least number of tokens (12,800 tokens in memory and 2,048 tokens in the generation context window). Similarly, in LongBook Event QA, M+ excels at identifying ground-truth events, showcasing its ability to reason over long-term dependencies. (2) Compared to Llama-3.1-3B-128k, the results on dataset Longbook-Event-QA suggest that tasks requiring reasoning capabilities benefit more from larger models with tailored structures for extended context windows rather than smaller models with longer context capacities. This highlights the importance of balancing model size and memory mechanisms under fixed GPU memory budgets. (3) Llama-3.1-8B-SnapKV underperforms Llama-3.1-8B-16k on LongBook-QA, indicating that solely relying on attention scores to select key tokens may not consistently yield optimal results. In contrast, M+ leverages a jointly trained retriever to identify and extract memory tokens, resulting in more effective performance on both datasets. (4) M+ outperforms Llama-3.1-8B with BM25 retriever, and Llama-3.1-8B with BM25 does not consistently outperform the original model Llama-3.1-8B-16k, particularly in tasks like Longbook-Event-QA which requires a more global understanding of the entire narrative. This highlights the limitations of chunk-level retrieval in scenarios that demand long-range comprehension. (5) Memory Efficiency: While M+ achieves state-of-the-art results, it does so with a highly memory-efficient design. Detailed memory cost comparisons are provided in Section 4.2.

### 4.2. GPU Cost Comparison

In this section, we report the maximum GPU memory allocated during the inference across both datasets for each method mentioned in Section 4.1. The results are shown in Table 1. From the results, we can find that M+ has the lowest GPU memory cost except for Llama-3.1-8B-16k. The reason that M+ uses fewer tokens but costs more GPU is that we have 12,800 tokens in each layer, while Llama-3.1-8B-16k has only one layer of 16k tokens. Therefore, we propose to offload the memory tokens on CPU, and reload them into GPU when the calculation reaches a certain layer. By "CPU offloading", we specifically mean offloading the memory vectors present in each layer of the model. It is important to note that other models, such as Llama-3.1-8B do not have memory vectors, so our CPU offloading can only be applied to MemoryLLM and M+. With that, we can sacrifice some I/O time cost but substantially decrease the GPU cost with-

*Table 1.* GPU Memory Cost Comparison. MemoryLLM-8B is the model obtained after Stage 1 training, serving as an ablation study.

| Method | GPU Memory Cost (MB) |
| --- | --- |
| Llama-3.1-8B-SnapKV | 32574.49 |
| Llama-3.2-3B-128k | 30422.70 |
| M+ | 21177.76 |
| Llama-3.1-8B-16k | 19239.21 |
| M+ (offload) | **17973.34** |
| MemoryLLM-8B | 21176.24 |
| MemoryLLM-8B (offload) | 17967.47 |

out affecting the performance. This leads to "M+ (offload)" which achieves the least GPU memory consumption. We also include the GPU cost of MemoryLLM-8B, which is the model obtained after Stage 1 described in Section 3.2.4. This shows that the Long-Term Memory does not incur more GPU costs.

### 4.3. Knowledge Retention Experiments

#### 4.3.1. EXPERIMENTAL SETTINGS

To evaluate the ability of M+ to recall long-term knowledge, we follow the experimental setup in MemoryLLM (Wang et al., 2024a) on datasets SQuAD and NaturalQA, formatted as (`context`, `question`, `answer`), where `context` and `question` are sentences, and `answer` is the correct response to the question. Consistent with Wang et al. (2024a), we extract samples with `answer` lengths of three tokens or fewer for SQuAD and four tokens or fewer for NaturalQA. After filtering out ambiguous examples that `gpt-4o-mini` fails to answer, we select the first 100 examples from the remaining answerable set to conduct our evaluation. To test the model's long-term retention ability, we insert distracting contexts between `context` and `question`. These distracting contexts are sampled from the training set of SQuAD. Both NaturalQA and SQuAD are constructed from Wikipedia and they are within the same domain. Moreover, the contexts in SQuAD training set are of more consistent lengths (around 300-500 tokens for each context), thus we sample the distracting contexts from SQuAD training set for both NaturalQA and SQuAD.

We compare with the following baselines: **MemoryLLM-7B**: The proposed model in Wang et al. (2024a), with the backbone Llama2-7B, and trained with C4 dataset. **Llama-3.1-8B-SnapKV**: We fix the cache size to 16384 and dynamically adjust the remaining key-value caches in the cache pool according to the newly injected distracting contexts (consistent with the settings from Section 4.1). The maximum prompt length is set to 49,152 (48k), which requires over 70 GB of GPU memory. We use longer prompt length here as we want to explore more knowledge retention abili-

Table 2. Experimental Results on LongBench.

|  | 2wikimqa | hotpotqa | qasper | musique | multifieldqa_en | narrativeqa | Avg |
|---|---|---|---|---|---|---|---|
| MemoryLLM-7B (20k) | 27.22 | 34.03 | 19.57 | 13.47 | 29.56 | 20.64 | 24.08 |
| Llama3.1-8B (8k) | 34.87 | 43.10 | 29.96 | 24.96 | 43.18 | 24.29 | 33.39 |
| Llama3.1-8B (16k) | 34.11 | 44.72 | 30.05 | 31.96 | 48.86 | 25.19 | 35.81 |
| M+ (8k) | 33.12 | 37.99 | 29.91 | 20.68 | 40.11 | 24.18 | 31.00 |
| M+ (16k) | 32.71 | 38.56 | 30.39 | 24.58 | 46.32 | 24.12 | 32.78 |

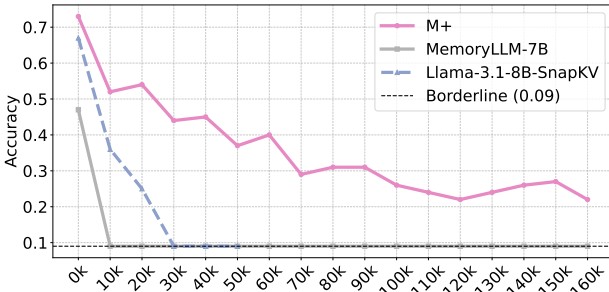

Figure 3. Knowledge Retention Results on SQuAD.

ties of Llama-3.1-8B-SnapKV.

### 4.3.2. EXPERIMENTAL RESULTS

The experimental results on SQuAD are presented in Figure 3. We present the results on NaturalQA in Appendix B.1 (Figure 8) as both figures have similar trends. From these figures, key observations include: (1) M+ significantly outperforms MemoryLLM-7B, demonstrating a substantial improvement in knowledge retention compared with the last version. (2) M+ surpasses Llama-3.1-8B equipped with SnapK, indicating that storing knowledge directly in memory is more effective than relying on key-value cache mechanisms. (3) Even though Llama-3.1-8B-SnapKV is given the context window 48k, it struggles to recall information injected more than 30k tokens earlier, highlighting the limitations of key-value cache methods for long-term knowledge retention.

### 4.4. Experimental Results on (Relatively) Short Documents

We evaluate the performance of M+ and Llama-3.1-8B on relatively short documents using the LongBench benchmark, considering input lengths of 8k and 16k tokens. The evaluation metric is QA-F1, following Bai et al. (2023). The results, presented in Table 2, show that M+ could match the performance of Llama-3.1-8B on 4 out of 6 datasets, except for hotpotqa and musique. This performance difference can be attributed to two primary factors: (1) **Random Dropping Mechanism**: M+ employs a random dropping mechanism that can lead to partial information loss. For instance, processing an 8k input requires splitting it into 12 chunks (each chunk being 512 tokens), while the last 2k tokens are directly included in the generation context

window. The first 6k tokens (12 chunks) are compressed into $256 + 256 * \frac{N-K}{N} + \cdots + 256 * (\frac{N-K}{N})^{11} = 2755.6$ tokens (with around 316.4 memory tokens dropped). For 16k tokens, the first 14k tokens are compressed 5530 tokens (with around 1638 tokens dropped). As some tokens are dropped, the performance may get affected slightly. Note that this could lead to a longer context window while sacrificing some of the performance in relatively shorter context tasks. (2) **Limited Cross-Chunk Attention**: When processing chunks into memory, M+ uses the last $K$ tokens and the hidden states from the new chunk as input (illustrated in Figure 1). In contrast, Llama-3.1-8B processes each chunk with access to all previous tokens, enabling cross-attention between chunks. While this approach allows Llama-3.1-8B to maintain full attention across chunks, it comes with significantly higher computational and memory costs due to the quadratic scaling of transformer computations. In comparison, M+ achieves linear computational scaling, making it more GPU-memory-efficient for processing extremely long inputs (see Section 4.2), albeit with some trade-off in performance in tasks with relatively shorter contexts.

### 4.5. Ablation Study

#### 4.5.1. ABLATION STUDY ON LONG-TERM MEMORY

In this section, we study the effectiveness of our long-term memory to ensure that the observed performance improvements over MemoryLLM-7B and Llama-3.1-8B-16k stem from the integration of LTM rather than solely from the additional training. Recall from Section 3.2.4 that the first two training stages do not use long-term memory. We compare with three models: (1) **MemoryLLM-8B**: The model obtained after Stage 1. It shares the same structure as MemoryLLM-7B (Wang et al., 2024a) but upgrades the backbone from Llama-2-7B to Llama-3.1-8B and includes changes such as dataset shifts and multi-LoRA settings. (2) **MemoryLLM-8B-Long**: The model obtained after Stage 2. (3) **M+**: The final model obtained after Stage 3.

**Long Context Modeling Ability Improves Over Stages**
We evaluate the three models on a held-out subset of Slim-Pajama containing 1000 examples with lengths between 32k and 64k tokens. We compute the validation loss for each model on this subset and report the results in Figure 4. The results demonstrate that long-context modeling abil-

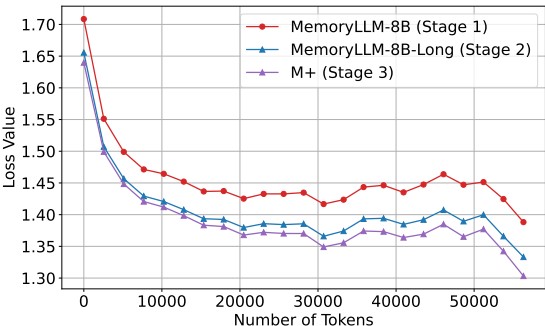

*Figure 4.* Validation loss comparison on a held-out subset from Slim-Pajama, consisting of 1,000 examples. The three models, MemoryLLM-8B, MemoryLLM-8B-Long, and M+, are obtained after Stages 1, 2, and 3, respectively (Section 3.2.4).

ity improves progressively across training stages, with M+ achieving the lowest validation loss, indicating the strongest performance on long-context inputs.

**Long-term memory Significantly Improves Knowledge Retention**   We further assess MemoryLLM-8B-Long and M+ on knowledge retention tasks using SQuAD and NaturalQA datasets with the same setting as in Section 4.3. In our experiments, we find MemoryLLM-8B-Long performs marginally better than MemoryLLM-8B on knowledge retention tasks, thus for simplicity, we omit the results of MemoryLLM-8B here. The results for MemoryLLM-8B-Long and M+ on dataset SQuAD are presented in Figure 5 and the results on NaturalQA are presented in Appendix B.2 (Figure 9). From the results, we can observe that (1) Despite having only 12,800 memory tokens, MemoryLLM-8B-Long retains knowledge for up to 30 tokens in NaturalQA and 50 tokens in SQuAD, demonstrating effective compression of information into memory tokens. (2) Stage 3 significantly enhances retention, extending the model's ability to recall knowledge from 50k to over 160 tokens. During inference, 2,560 tokens are retrieved from long-term memory, combined with 10,240 tokens from short-term memory, resulting in 12,800 effective memory tokens. These results underscore the effectiveness of our long-term memory mechanism in improving knowledge retention and handling extremely long contexts.

**Long-term memory does not affect the performance on relatively short documents**   To show whether long-term memory affects the model's performances on relatively short documents, we conduct ablation study with models from three different stages on the dataset LongBench while fixing the context window as 8k. The results are shown in Table 3. From the table we can see that M+ has similar performance as MemoryLLM-8B-Long on 8k context window. This means adding long-term memory does not affect the performance on relatively short documents. Meanwhile, MemoryLLM-8B-Long is significantly better than

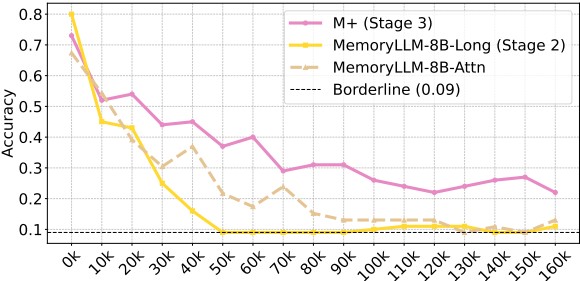

*Figure 5.* Ablation Study on SQuAD dataset.

MemoryLLM-8B, which can be attributed to the inclusion of longer training examples in Stage 2 (4k–64k), whereas `fineweb-edu` (used in Stage 1) contains very few examples longer than 4k.

### 4.5.2. ABLATION STUDY ON RETRIEVER

We conduct an ablation study to evaluate the performance of our trained retriever compared to an attention-based retrieval method. Using the SQuAD and NaturalQA datasets, we compare M+ with an attention-based retrieval method inspired by H2O (Zhang et al., 2023). In the attention-based method (M+-Attn), the key-value cache of past tokens is stored, and during generation, a fixed number of tokens are retrieved from the cached keys and values based on their attention scores. To match our approach, we use the same short-term memory as M+, but the memory tokens in the long-term memory are retrieved according to the attention scores rather than using our retriever. To implement this method, we adapt M+ to store key-value caches instead of hidden states in the long-term memory to avoid any additional computation cost. During generation, for each token, we extract 2,560 keys and values for each head from the long-term memory, along with the 10,240 memory tokens in the current memory pool. The results on SQuAD are shown in Figures 5 and the results on NaturalQA are shown in Appendix B.2 (Figure 9). From the figures we can see that M+ substantially outperforms M+-Attn, showing the advantages of our trained retriever over the attention-based approach in terms of knowledge retention and retrieval efficiency.

### 4.6. Analysis

#### 4.6.1. MODEL QUALITY WITHIN CONTEXT WINDOW

M+ uses 12,800 memory tokens alongside a 2,048-token generation context window. In this section, we evaluate the model's performance within the standard 2,048-token context window to ensure that the addition of memory does not degrade its base capability. We randomly select 1,000 examples from the `fineweb-edu` dataset (snapshot `CC-MAIN-2024-10`), which does not overlap with the training data. For this evaluation, we cap the input sequence length at 2,048 tokens and report perplexity for both M+ and LLaMA-3.1-8B. The results show that LLaMA-3.1-8B achieves a perplexity of 1.9734, while M+ records a sim-

*Table 3.* Ablation study of the effects of different stages on LongBench.

|  | **2wikimqa** | **hotpotqa** | **qasper** | **musique** | **multifieldqa_en** | **narrativeqa** | Avg |
|---|---|---|---|---|---|---|---|
| MemoryLLM-8B (Stage 1, 8k) | 32.30 | 33.39 | 23.88 | 12.37 | 35.91 | 21.46 | 26.55 |
| MemoryLLM-8B-Long (Stage 2, 8k) | 32.23 | 37.86 | 31.62 | 20.35 | 42.16 | 23.49 | 31.29 |
| M+ (Stage 3, 8k) | 33.12 | 37.99 | 29.91 | 20.68 | 40.11 | 24.18 | 31.00 |

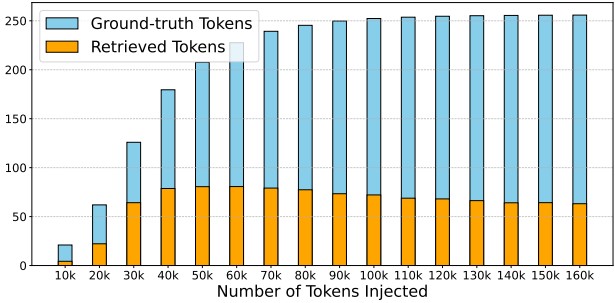

*Figure 6.* Number of ground-truth tokens in long-term memory and the number of retrieved groud-truth tokens as more tokens are injected into the memory.

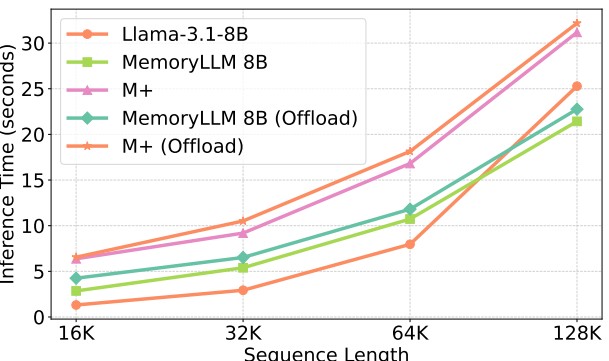

*Figure 7.* Latency Analysis

ilar perplexity of 1.9828. These findings indicate that M+ maintains competitive performance on documents shorter than 2,048 tokens, confirming that the base model's quality within the context window remains intact.

### 4.6.2. RETRIEVAL QUALITY

In our implementation, the long-term memory is initially of size 5120, and then it gradually increases to 80k in our knowledge retention experiments (it hits 81,276 when there 160k tokens are injected). To access retrieval quality, we leverage the knowledge retention task with SQuAD dataset, where the first $K = 256$ tokens are critical for answering the questions. These $K = 256$ tokens are denoted as ground-truth tokens. We track the number of ground-truth tokens in the long-term memory and how many tokens are retrieved back into the "Extracted LTM" pool in Figure 1 when queried after various numbers of tokens are injected. We present the results in Figure 6, demonstrating the retrieval quality as more tokens are dropped from the memory pool to the long-term memory. From the figure we can see that around 30% tokens are retrieved. For reference, random retrieval would retrieve $2,560/81,276 = 3\%$ tokens.

### 4.6.3. LATENCY ANALYSIS

While M+ introduces additional computation due to the memory token retrieval from the long-term memory, we perform a detailed analysis to quantify this latency. Specifically, we analyze latency under the setting of a 128k input. For reference, we use the processing time of Llama-3.1-8B performing a forward pass on 131,071 (=128k-1) tokens to generate the final token. To ensure fairness, we inject 131,072 - 2,048 tokens into the memory and ask M+ to predict the last token using the remaining 2,047 tokens. We

focus on the following settings: (1) Llama-3.1-8B-128k. To analyze the latency, we use Llama-3.1-8B with a full context window 128k; (2) MemoryLLM-8B (After Stage 1); (3) M+ (After Stage 3); (4) MemoryLLM-8B (Offload): we offload the memory onto CPU and load the corresponding memory tokens into GPU when the computation hits a certain layer; (5) M+ (offload): Offloading the memory onto CPU and load them back when necessary. All experiments in this section are conducted on a single H100 GPU. The results are shown in Figure 7. From the figure, we could find that (1) MemoryLLM-8B has slightly higher latency than Llama-3.1-8B in relatively shorter documents (16k, 32k, 64k) but has lower latency on long documents (128k); (2) M+ has higher latency than MemoryLLM-8B, where the latency is mainly introduced by the retrieval process. (3) Offloading the memory onto CPU introduces slightly more latency, while it becomes negligible when the sequence grows longer. In the case of 128k input, the introduced latency for M+ (offload) compared with M+ is 1 second, leading to 3% additional computation time for M+.

## 5. Conclusion and Future Work

In this work, we present M+, an enhanced memory-augmented language model that extends the long-term retention abilities of MemoryLLM. By integrating a long-term memory (LTM) mechanism with a co-trained retriever, M+ effectively retrieves and utilizes past information, significantly extending the knowledge retention abilities from MemoryLLM, achieve superior performances in long-context understanding tasks compared with recent baselines given the similar budget of GPU memory. In future work, we plan to reduce CPU-GPU communication overhead, enabling more efficient generation with M+.

## Impact Statement

This work introduces a memory-augmented approach for Large Language Models (LLMs), enabling them to more effectively retain and retrieve long-term information and thereby offering potential benefits in areas such as education, research, and industry. The increased memory capacity could potentially raise concerns regarding AI safety, reliability, and fairness. If not carefully managed, these models could propagate biased content over extended text spans or store sensitive information for unintended durations. It is therefore crucial to employ robust safeguards, including bias mitigation strategies and ongoing oversight, to prevent misuse or the reinforcement of harmful content. Beyond considerations already inherent to LLMs, we do not foresee other significant societal impacts arising from this work.

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

# A. Justifications of using `deepspeed-stage-2`

Eight A100 GPUs support the following configurations:

- Full fine-tuning with an 8k context window using Fully Sharded Data Parallel (FSDP).

- 6k context window with full attention using `deepspeed-stage-2`.

- 32k context window with full attention using `accelerate` and `deepspeed-stage-3-offload`. However, saving models in this configuration encountered version incompatibility issues and we haven't found solutions online.

Based on these trails, we do not scale up the model with `deepspeed-stage-3-offload` or FSDP, but choose to use `deepspeed-stage-2` and set the cross-attention to be of the shape 2048 by 14848.

# B. Experiments on datasets NaturalQA

### B.1. Knowledge Retention Experiments on NaturalQA

The results of knowledge retention experiments on NaturalQA are shown in Figure 8.

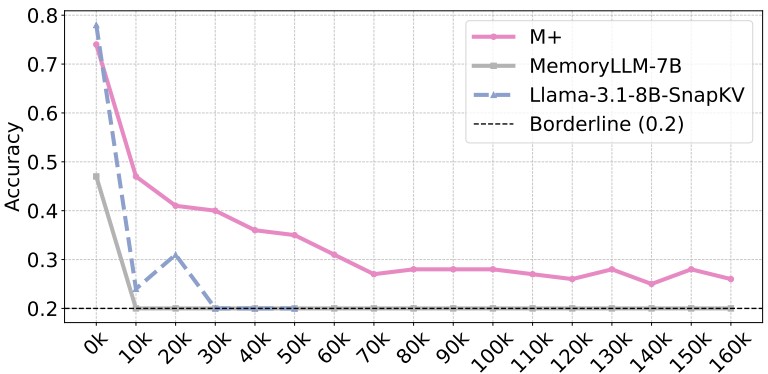

*Figure 8.* Knowledge Retention Results on NaturalQA.

### B.2. Ablation Study on NaturalQA

The results of ablation study on NaturalQA are shown in Figure 9.

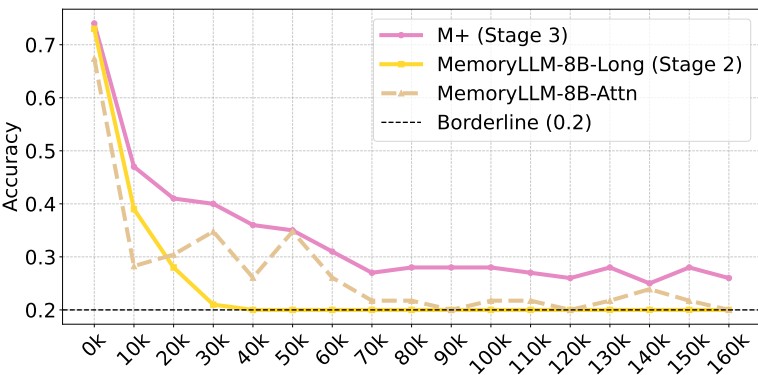

*Figure 9.* Ablation Study on NaturalQA dataset.

## C. Statistics of the Dataset of Long Documents

We go through the whole dataset SlimPajama-627B and extract all dataset that have more than 4k tokens using the tokenizer of Llama-3.1-8B. The statistics are shown in Table 4. We show six categories here (4k-8k, 8k-16k,16k-32k,32k-64k,64k-128k,128k+) but we only use the data within the first four categories (4k-8k, 8k-16k,16k-32k,32k-64k). This is because the examples longer than 64k are mainly from the category **Book** and lack diversity.

| Range | Total | CommonCrawl | GitHub | ArXiv | C4 | StackExch. | Wikipedia | Book |
|---|---|---|---|---|---|---|---|---|
| **4k–8k** | 11,189,999 | 7,759,741 (69.35%) | 692,224 (6.19%) | 286,537 (2.56%) | 1,825,018 (16.31%) | 142,457 (1.27%) | 481,854 (4.31%) | 2,168 (0.02%) |
| **8k–16k** | 4,706,687 | 3,273,619 (69.55%) | 270,369 (5.74%) | 550,192 (11.69%) | 439,143 (9.33%) | 20,284 (0.43%) | 146,545 (3.11%) | 6,535 (0.14%) |
| **16k–32k** | 1,607,064 | 968,714 (60.28%) | 95,445 (5.94%) | 423,401 (26.35%) | 70,223 (4.37%) | 1,510 (0.09%) | 34,323 (2.14%) | 13,448 (0.84%) |
| **32k–64k** | 443,438 | 224,168 (50.55%) | 32,653 (7.36%) | 146,582 (33.06%) | 3,413 (0.77%) | 102 (0.02%) | 5,940 (1.34%) | 30,580 (6.90%) |
| **64k–128k** | 192,515 | 72,583 (37.70%) | 11,753 (6.10%) | 27,942 (14.51%) | 38 (0.02%) | 5 (0.00%) | 507 (0.26%) | 79,687 (41.39%) |
| **128k+** | 98,097 | 23,721 (24.18%) | 4,523 (4.61%) | 5,167 (5.27%) | 0 (0.00%) | 2 (0.00%) | 49 (0.05%) | 64,635 (65.89%) |

*Table 4.* Number of examples by sequence-length range and source (counts and percentages).

## D. Additional Training Details

In our training, we follow MemoryLLM (Wang et al., 2024a) and design three sub-tasks:

- **Two-Chunk Training**: Given a document split into two chunks $(x_1, x_2)$, we inject $x_1$ into the memory and update the transformer $\phi$ using the loss on $x_2$. Notably, we retain the gradients across both forward passes.

- **Multi-Chunk Training**: For documents with multiple chunks $(x_1, \ldots, x_n)$, we inject $x_1, \ldots, x_{n-1}$ into the memory while detaching gradients, then update $\phi$ using the loss on $x_n$.

- **Revisiting Cached Chunks**: Since the memory is continually updated during training, we cache the last chunk $x_n$ of earlier documents and revisit it periodically. When revisiting $x_n$, there are already many chunks injected between $x_1, \cdots, x_{n-1}$ and $x_n$. We denote the number of injected chunks between $x_{n-1}$ and $x_n$ as *revisit distance*. We carefully tune the probability of deleting and updating the cache after each training step, and we manage to maintain the average revisit distance to be around 60 for Stage 1 & Stage 2, and maintain the average distance to be around 200 for Stage 3.

## E. Discussions

### E.1. Similarities to Attention-Based Retrieval Methods

In M+, we use a co-trained retriever to retrieve the hidden states. In this process, we acknowledge that our method shares some similarities with prior approaches that use attention to retrieve keys and values. However, there are critical differences that make our approach unique and practically advantageous:

- **Efficiency**: Methods such as SnapKV maintain and retrieve key-value pairs per head, which becomes extremely costly when scaled. In our setting—with 32 layers and 32 attention heads per layer—this requires 1024 retrievals per query, resulting in significant latency (as noted in line 59 of our paper). In contrast, M+ uses a co-trained retriever to retrieve memory tokens, which are compressed hidden states. This results in only 32 total retrievals—one per layer—dramatically reducing both computational cost and latency.

- **Performance**: In Figure 6, the curve labeled MemoryLLM-8B-Attn follows the SnapKV-style approach of retrieving key-value pairs using attention per head. As shown in the figure, it performs substantially worse than M+, highlighting that our co-trained retriever not only improves efficiency but also yields better results in practice compared with attention-based retrievals.

- **Design**: Note that our training setup includes both relevant and irrelevant documents (See details in Appendix D), making it well-suited for contrastive learning. This allows us to effectively train the retriever, which integrates naturally into our overall training framework.

### E.2. The Form of Long-Term Memory (Hidden States vs. KV)

In our work, we choose the use hidden states as the latent-space memory instead of key-value (KV) caches. This is based on the following two considerations:

- **Compression Efficiency**: As detailed in the paper, we compress each 512-token chunk into 256 memory vectors per layer in a lossless manner. In contrast, KV-based methods often require downsampling—e.g., dropping half the keys and values—to control memory size, resulting in unavoidable information loss.

- **Retrieval Efficiency and Performance**: As described above, hidden states can be effectively retrieved using our co-trained retriever, requiring only 32 retrievals for each query. In contrast, a KV-cache approach would demand up to 1024 retrievals, significantly increasing computational cost. Furthermore, as shown in Figure 6, using hidden states yields better performance compared to using KV caches.

### E.3. Latency and Memory Consumption while Scaling

We aim to discuss the scalability of M+ by analysing the latency and memory consumption when scaling up. Theoretically, the end-to-end *retrieval latency* scales linearly with three key variables:

(1) Hidden size of the retriever, denoted by $d$. In M+, we set $d = 256$, whereas the base model uses $d = 4096$.

(2) Size of long-term memory, denoted by $s$. We cap this at 150k entries.

(3) Number of transformer layers, denoted by $L$. For LLAMA-3-8B, $L = 32$.

Hence,

$$\text{latency} \propto d\, s\, L.$$

Because we hold the long-term memory size $s$ fixed when scaling the model, $s$ is effectively a constant:

$$\text{latency} \propto d\, L.$$

Both $d$ and $L$ grow with the model size $M$, following

$$M \propto d\, L,$$

which implies a **linear relationship** between retrieval latency and model size:

$$\text{latency} \propto M.$$

**Concrete Example**  Scaling from LLAMA-3-8B ($d = 4096$, $L = 32$) to LLAMA-3-70B ($d = 8192$, $L = 80$) yields

$$\frac{8192 \times 80}{4096 \times 32} = 5,$$

i.e. a $\sim 5\times$ increase in retrieval latency. For comparison, the parameter count rises by $\frac{70\text{B}}{8\text{B}} \approx 8.75\times$, showing that latency scales roughly linearly—rather than quadratically—with model size.

**Memory Consumption**  The extra *memory overhead* from our method arises solely from the introduced memory tokens. This overhead also scales linearly with both $d$ and $L$; thus, the move from LLAMA-3-8B to LLAMA-3-70B incurs an analogous $\sim 5\times$ increase in memory usage, mirroring the latency scaling.

## E.4. FLOPs Comparison

We report the total FLOPs for generating one token after processing a sequence of varying lengths (from 2k to 128k), using a single H100 GPU. We employ the `torch.profiler` library to capture FLOPs during inference. The results are as follows:

| Sequence Length | LLaMA-3.1-8B | M+ |
|:---:|:---:|:---:|
| 2048 | $5.68 \times 10^{13}$ | $6.92 \times 10^{13}$ |
| 4096 | $1.13 \times 10^{14}$ | $1.32 \times 10^{14}$ |
| 8192 | $2.26 \times 10^{14}$ | $2.55 \times 10^{14}$ |
| 16384 | $4.48 \times 10^{14}$ | $5.01 \times 10^{14}$ |
| 32768 | $8.88 \times 10^{14}$ | $9.86 \times 10^{14}$ |
| 65536 | $1.75 \times 10^{15}$ | $1.94 \times 10^{15}$ |
| 131072 | `OOM` | $3.78 \times 10^{15}$ |

From the results, we observe that M+ and LLaMA-3.1-8B exhibit comparable FLOPs across all tested sequence lengths. Notably, while LLaMA-3.1-8B runs out of memory (`OOM`) at the 128k setting, M+ remains functional, highlighting its superior scalability for long-context inference.

## E.5. Interpretability of Memory Vectors

Our memory vectors can be viewed as hidden states within the transformer layers, with the key difference being that they may store more **compressed** information due to their persistent role across sequences. As such, the type of information they capture should be similar to the representations observed in the intermediate layers of a transformer when processing text.

Across layers, we hypothesize that the memory vectors follow a similar pattern to what has been reported in prior work on transformer interpretability (Jawahar et al., 2019; Simoulin & Crabbé, 2021):

- **Lower layers** tend to encode more **surface-level features**,

- **Higher layers** tend to encode more **semantic or abstract information**.

Regarding long-term memory, it is constructed by **randomly dropping** vectors from the short-term memory and storing them for extended use. Importantly, long-term memory vectors are structurally **identical** to short-term ones. This means that, at any point, *swapping a vector between long-term and short-term memory has no immediate effect* on model behavior.

In essence, the long-term memory acts as a **cache** that helps memory vectors **persist over time** rather than being overwritten too quickly.

