# OpenReview forum: "M+: Extending MemoryLLM with Scalable Long-Term Memory"
_ICML.cc/2025/Conference — ICML 2025 poster_

### Official Review · Reviewer_zEbf · 2025-02-21

**Overall Recommendation:** 3

**Summary:**

The paper proposes an LLM memory-augmentation method called M+. By building on MemoryLLM, it improves the long-context understanding and information retention of a base LLM. They introduce what they call long-term memory vectors that are extracted by a trained retriever. M+ outperforms the base model and other existing baselines on long-context QA and knowledge retention benchmarks.

**Claims And Evidence:**

M+ improves the long-context understanding and information retention of LLMs. This has been validated by the experiments in section 4.1 and 4.3. M+ also claims to be an efficient method but this is not clearly supported by the evidence presented in the paper.  It has higher memory consumption and latency than the original base model. Adding CPU offloading to Table 1 is a bit misleading. It could be applied to any model and it is not specific to M+.

**Essential References Not Discussed:**

Reccurrent Memory Transformers and their variants (https://proceedings.neurips.cc/paper_files/paper/2022/hash/47e288629a6996a17ce50b90a056a0e1-Abstract-Conference.html
, https://arxiv.org/abs/2304.11062) were not discussed in the paper. They also study latent-space memory.
The MemoryPrompt (https://aclanthology.org/2024.lrec-main.976/) paper analyzes the content of these memory vectors.

**Experimental Designs Or Analyses:**

Concerning analysis, not much is said about the memory vectors. It would greatly help the community to study what kind of information is contained in the memory vectors.

**Methods And Evaluation Criteria:**

The benchmark datasets used in the paper are appropriate as they evaluate the long-context memory capabilities of models. However, the models were not evaluated on standard long-context benchmarks like the classic Needle-in-a-Haystack test and LongBench.

**Other Comments Or Suggestions:**

Figure 3 is a bit hard to read. The scales on the 2 sides are different but combined in the same figure.
A Limitations section would help better delineate the actual contributions of the paper.

**Other Strengths And Weaknesses:**

None

**Questions For Authors:**

1- What is the performance on the following standard long-context benchmarks: Needle-in-a-Haystack test and LongBench?

2- What is the memory consumption of MemoryLLM in your experiments?

3- Is the performance of the base Language Model affected by the introduction of the long-term latent-space memory? (perplexity and other metrics on standard LM benchmarks)

4- What is the performance of a simple RAG baseline?

5- Can you provide an estimate of how the memory consumption and latency would behave for very big models?

6- FLOPs was never reported. What is the FLOPs compared to the base model and other baselines? What are the results if you do FLOP-matched comparisons?

7- From an interpretability perspective, what kind information is contained in these memory vectors? How do they differ across layers? What distinguishes the long-term memory vectors from the short-term ones?

**Relation To Broader Scientific Literature:**

The paper studies the long-context extension of LLMs. It also builds on an existing approach called MemoryLLM.

**Theoretical Claims:**

None

---

> ### Author Rebuttal · Authors · 2025-04-01
>
> **Claims And Evidence:**
>
> We would like to clarify that by “CPU offloading,” we specifically mean **offloading the memory vectors** present in each layer of the model. In our setup, each layer contains 12,800 memory vectors, and it is unnecessary to keep all of them simultaneously in GPU memory. Instead, we can store them on the CPU and load them into GPU memory only when the corresponding layer is being computed. **It is important to note that other models such as LLaMA 3.1–8B do not have memory vectors, so our CPU offloading can only be applied on MemoryLLM and M+**. We will add more clarifications.
>
> **Supplementary Material:**
> We will publish the code and model upon acceptance and make sure of the reproducibility.
>
> **Essential References Not Discussed:**
> Thank you for bringing these up, we will add these works ([1], [2], [3]) into our paper.
>
> **Comments, Suggestions and Questions:**
> **Figure 3**: We will revise the scales to ensure consistency across subplots.
> **Limitation Section:**: We briefly mentioned our plan to reduce CPU-GPU communication overhead in future work, we will add a section to discuss this.
>
> **[Q1]**
>
> (1) **Evaluations on LongBench**: We would like to note that our evaluations on LongBench are included in Section 4.4.
>
> (2) **Experimental Results on NIAH**: We would like to emphasize two key points:
> - **The scope of this paper goes beyond NIAH**: Our primary goal is to capture and reason over global information, rather than focusing solely on retrieval. Thus we focus on on longbook question answering and introduce a new dataset, Longbook-Event-QA (Section 4.1.1), which requires the model to understand a broader part of the book. In contrast, NIAH primarily evaluates retrieval ability, regardless of whether the model comprehends the broader story.
> - **Our knowledge retention experiments share similarities with NIAH**: In Section 4.5 (Figures 5 and 6), the model is tasked with recalling information from a context it encountered long ago and answering related questions. This setup resembles document-level retrieval and aligns conceptually with the goals of NIAH, suggesting that our work addresses similar challenges from a different perspective.
>
> **[Q2]**
>
> We add the following two rows to Table 1 to report the GPU memory consumption of MemoryLLM: MemoryLLM-8B, 21176.24MB; MemoryLLM-8B (offload): 17967.47MB. These results show that MemoryLLM-8B has comparable GPU memory usage to M+.
>
> **[Q3]**
>
> On 1,000 unseen examples from Fineweb-Edu (2048-token limit), M+ achieves a perplexity of 1.9828 vs. 1.9734 for LLaMA-3.1-8B, showing no degradation in base model performance.
>
> **[Q4]**
>
> Using BM25-based RAG (retrieving up to 4 chunks of 4,096 tokens), we observe limited gains: LLaMA-3.1-8B + BM25 achieves 0.1623 on LongbookQA and 0.2065 on LongBook-Event-QA, slightly improving over LLaMA-3.1-8B-16k (0.1514 / 0.2362) on one task but worse on the other. In contrast, M+ achieves the best scores: 0.1755 / 0.2470. This shows RAG offers no consistent benefit and may hurt performance when global context is needed.
>
> **[Q5]**
>
> Retrieval latency scales as $\text{latency} \propto d_r \cdot s \cdot L \propto d \cdot s \cdot L$, where $d_r$ is the retriever hidden size ($d/20$), $s$ is the memory size (maximum 150k, independent of model size), and $L$ is the number of layers. Since $s$ is constant, latency simplifies to $\text{latency} \propto d \cdot L$. Given $M \propto d \cdot L$, we have $\text{latency} \propto M$. As for the memory consumption, Memory vectors overhead also scales linearly with $d$ and $L$.
>
> **[Q6]**
>
> M+ and LLaMA-3.1-8B have similar FLOPs (e.g., 6.92e13 vs. 5.68e13 at 2k, 1.94e15 vs. 1.75e15 at 64k). At 128k, LLaMA-3.1-8B runs out of memory, while M+ runs property and has the total FLOPs as 3.78e15.
>
> **[Q7]**
>
> Our memory vectors can be viewed as hidden states within the transformer layers, with miner differences being that they may store more **compressed** information. Thus the information they capture should be similar to the representations seen in the intermediate layers of a transformer when processing text.
> Across layers, we hypothesize that the memory vectors follow a similar pattern to what has been observed in prior work on transformer interpretability [4, 5]:
> - **Lower layers** tend to encode more **surface-level features**,
> - **Higher layers** tend to encode more **semantic or abstract information**.
>
> Regarding long-term memory, it is constructed by **randomly dropped** vectors from the short-term memory and storing them for extended use. Importantly, long-term memory vectors are structurally **identical** to short-term ones.
>
> References:
>
> [1] Recurrent Memory Transformer
> [2] Scaling Transformer to 1M tokens and beyond with RMT
> [3] MemoryPrompt: A Light Wrapper to Improve Context Tracking in Pre-trained Language Models
> [4] How Many Layers and Why? An Analysis of the Model Depth in Transformers
> [5] What does BERT learn about the structure of language?

---

> > ### Comment · Reviewer_zEbf · 2025-04-02
> >
> > The authors have helped clarify all the ambiguous aspects of the paper. They also provided detailed answer to almost all the questions. I will update my score accordingly. It is important to include all the details in the paper for better clarity and reproducibility. However, the paper still lacks a proper quantitative analysis of the memory vectors from an interpretability perspective. The answer to Q7 are only speculations that are not properly substantiated.

---

> > > ### Author Response · Authors · 2025-04-03
> > >
> > > We sincerely thank the reviewer for raising the score and are extremely grateful for their constructive feedback. We will definitely include all relevant details and clarifications in the final version of the paper.
> > >
> > > As for the concerns regarding interpretability, we acknowledge that the current version lacks a thorough interpretability analysis of M+ and we sincerely thank the reviewer for bringing this up. We would like to provide some more clarifications here: Our primary focus in this work is on exploring the model structures and demonstrating the model’s performance. We would like to respectfully note that this approach aligns with the trajectory of several influential works in the field. For example, seminal papers such as BERT: Pre-training of Deep Bidirectional Transformers for Language Understanding [1] and Attention Is All You Need [2] initially prioritized performance and introduced novel architectures, with interpretability analyses and theoretical insights following in subsequent research and other papers.
> > >
> > > We hope this “performance-first, analysis-later” approach can be seen as a valid path for impactful contributions, and we fully intend to explore the interpretability of M+ in future work.
> > >
> > > [1] BERT: Pre-training of Deep Bidirectional Transformers for Language Understanding
> > > [2] Attention Is All You Need

---

### Official Review · Reviewer_v8Qd · 2025-03-09

**Overall Recommendation:** 4

**Summary:**

This paper presents M+, an enhanced memory-augmented language model that extends long-term memory retention beyond conventional limits. Building on MemoryLLM, M+ integrates long-term memory with a co-trained retriever.  Extensive experiments across long-context understanding, question answering, and knowledge retention benchmarks demonstrate that M+ consistently outperforms prior baselines, offering a robust and efficient solution for processing extremely long documents.

## update after rebuttal
Most of my concerns are addressed so I have raised my reviewing score.

**Claims And Evidence:**

Yes. Most of the claims are supported by a range of quantitative experiments and ablation studies. For example, the paper demonstrates through long-book QA, Event QA and knowledge retention benchmarks that M+ outperforms baselines like MemoryLLM and Llama families. Detailed GPU memory cost comparisons and latency analysis further back the claim that M+ achieves extended retention with efficient resource usages.

**Essential References Not Discussed:**

No

**Experimental Designs Or Analyses:**

Yes. The knowledge retention experiments and ablation studies are well suited to testing M+’s ability to remember and retrieve distant information.
While the paper’s approach to evaluating memory retention is well designed, the evaluation dataset and task types are limited. More complex datasets or practical applications are needed to demonstrate the effectiveness of the method [1, 2].
[1] Maharana, Adyasha, et al. "Evaluating Very Long-Term Conversational Memory of LLM Agents." Proceedings of the 62nd Annual Meeting of the Association for Computational Linguistics (Volume 1: Long Papers). 2024.
[2] Lu, Junru, et al. "Memochat: Tuning llms to use memos for consistent long-range open-domain conversation." arXiv preprint arXiv:2308.08239 (2023).

**Methods And Evaluation Criteria:**

Yes. The propose method M+ for integrating a long-term memory module into MemoryLLM match the stated goal of extending context retention beyond 20k tokens, and the chosen evaluation benchmarks (e.g., LongBook-QA, Event QA, and knowledge retention tasks) effectively measure whether the model can recall and use information from far in the past.

**Other Comments Or Suggestions:**

1.	In the comparison experiments (Sections 4.1 and 4.2), since M+ is largely based on MemoryLLM, it would be beneficial to include direct comparisons showing how much M+ actually improves over MemoryLLM. Observing that margin explicitly could clarify the advantages of the proposed enhancements.
2.	In Table 2, the authors report results on shorter-document tasks, yet M+ still outperforms MemoryLLM by a noticeable margin. Given that M+’s primary benefit appears to be in handling extremely long contexts, it would be helpful to clarify why it achieves such gains even with relatively small datasets.

**Other Strengths And Weaknesses:**

The core ideas of this paper build on existing latent-space memory approaches and chunk-based processing techniques. While the practical enhancements (e.g., CPU offloading to keep GPU usage low and adding a co-trained retriever for selective recall) are valuable, these methods can be viewed as incremental refinements, and I found them limited in originality and novelty, despite the paper’s clear demonstration of utility.

**Questions For Authors:**

No

**Relation To Broader Scientific Literature:**

The paper builds on research in memory-augmented language models, particularly latent-space memory approaches (e.g., MemoryLLM) that store and retrieve hidden-state representations instead of raw tokens. The authors go a step further by co-training a retriever with their model, in contrast to attention-based retrieval from key-value caches. Their method also connects with work on long-context modeling by pushing context windows into the 100k+ range. By demonstrating improved knowledge retention and efficient retrieval on challenging long-QA tasks, the paper contributes to broader efforts of making LLMs handle truly extensive contexts while remaining computationally feasible.

**Theoretical Claims:**

The paper primarily focuses on an empirical framework rather than on detailed proofs of novel theorems.

---

> ### Author Rebuttal · Authors · 2025-04-01
>
> We are sincerely grateful for Reviewer v8Qd’s recognition of our work. Below, we address the reviewer’s questions in detail:
>
> **[Q1] Direct Comparison between M+ and MemoryLLM: ** Thank you for raising this important point. In developing M+, we incorporated a number of significant improvements over MemoryLLM-7B, including:  (1) a multi-LoRA architecture, (2) refined dataset selection (Fineweb-Edu and SlimPajama), and (3) a carefully designed data curriculum. These enhancements led to substantial performance gains, and as a result, we consider M+ to be in a different performance tier compared to MemoryLLM-7B.
>
> That said, to provide a fair and controlled comparison of the architectural differences—specifically the benefits introduced by M+ – we retrained the original MemoryLLM with: (1) The LLaMA3-8B backbone; (2) The same multi-LoRA design; (3) More advanced datasets (Fineweb-Edu and SlimPajama). The retrained versions are included in our ablation study in Section 4.5, where we have two variants:
>
> MemoryLLM-8B: We switch the backbone to Llama-3.1-8B and use multi-LoRA design and Fineweb-Edu dataset to continually train the model. This is exactly Stage 1 mentioned in Section 3.2.4.
>
> MemoryLLM-8B-Long: After obtaining MemoryLLM-8B, we apply Stage 2 with long documents extracted from SlimPajama to enhance the model’s long document understanding abilities.
>
> Although this section is titled “Ablation Study,” it effectively serves as a direct comparison between the MemoryLLM architecture and M+, isolating the effect of long-term memory while controlling for backbone model and training data. The comparison includes the following aspects:
>
> 1. **Perplexity (Figure 5):** M+ demonstrates lower perplexity than both MemoryLLM-8B and MemoryLLM-8B-Long. The performance gap between M+ and MemoryLLM-8B-Long is smaller, reflecting the benefits of the long-input training in both models.
>
> 2. **Knowledge Retention (Figure 6):** M+ shows a significant advantage over MemoryLLM-8B-Long in retention tasks. Notably, we observed that MemoryLLM-8B and MemoryLLM-8B-Long perform similarly on this task, underscoring the importance of the architectural changes in M+.
>
> 3. **Performance on Relatively Short Documents:** To further address your question, we have conducted additional experiments on shorter documents (8k tokens), and we will include these results in the paper. The results are as follows:
>
>     | Model                   | 2wikimqa | hotpotqa | qasper | musique | multifieldqa\_en | narrativeqa | Avg   |
>     |------------------------|----------|----------|--------|---------|---------------|----|--------------|
>     | MemoryLLM-8B (8k)      | 32.30    | 33.39    | 23.88  | 12.37   | 35.91         | 21.46 | 26.55  |
>     | MemoryLLM-8B-Long (8k) | 32.23    | 37.86    | 31.62  | 20.35   | 42.16         | 23.49 | 31.29  |
>     | M+ (8k)                | 33.12    | 37.99    | 29.91  | 20.68   | 40.11         | 24.18 | 31.00  |
>
> These results show that while M+ and MemoryLLM-8B-Long perform similarly on shorter documents (as expected), M+ provides significant gains on long-context tasks (as shown in Figure 6). This further supports our claim that the long-term memory architecture in M+ is beneficial primarily for extended contexts. Additionally, the improved performance of MemoryLLM-8B-Long over MemoryLLM-8B on short documents can be attributed to the inclusion of longer training examples in Stage 2 (4k–64k), whereas Fineweb-Edu (used in Stage 1) contains very few examples longer than 4k.
>
> In summary, **Section 4.5 provides a direct and controlled comparison between M+ and MemoryLLM**, where the primary architectural difference is the inclusion of long-term memory. We will explicitly clarify this point in the paper and include the new short-document comparison results for completeness.

---

### Official Review · Reviewer_f5Uz · 2025-03-12

**Overall Recommendation:** 3

**Summary:**

Memory model: memory pool (based on MemoryLLM) and a long-term memory with additional temporal information. Every time, the memory pool is updated, and a subset of tokens is dropped to the Long-term memory. For recall from memory, a small subset of the long-term memory vectors is retrieved according to the dot product with the query from input. The search in the long-term memory uses a low dimensional projection of the vectors (keys) and the input query for such recall. These projectors are trained separately as part of the retrieval mechanism for the long-term memory. The M+ model is fine-tuned with 2 sets of weights based on LoRA: for reading and for updating the memory. Also, it was trained in 3 stages. The first stage fine-tunes Llama-3.1-8B following the MemoryLLM setup to incorporate the memory pool. This process is done on shorter documents. The second stage extends this to a balanced set of documents of various lengths. The last stage introduces the long-term memory and adapts the model to it on a new subset of long context documents. The M+ model is then evaluated on the LongBook-QA, a synthetically generated extraction of events called LongBookEvent-QA, SQuAD, NaturalQA, and LongBench. The baselines are Llama3.1-8B-16k, a similar version with SnapKV, and a Llama3.1 3B model with 128k context length. The results show that M+ has higher performance with lower or comparable memory consumption. The authors include ablation study comparing M+ after each fine-tuning stage for validation loss convergence and knowledge retention (SQuAD and NaturalQA).

## update after rebuttal

I appreciate the authors replying to my questions. There is no additional evidence that leads me to update my score.

**Claims And Evidence:**

The claims made in this work seem appropriately supported with convincing evidence.

**Essential References Not Discussed:**

To the best of my knowledge the authors cite many previous work on memory-augmented LLMs. They appropriately introduce the predecessor MemoryLLM.

**Experimental Designs Or Analyses:**

Mostly, the design and the analyses look sound. I would encourage the authors to test on datasets that evaluate for long context, instead of selecting documents from a general dataset. The intention behind the ablation study is valuable, however, it is not evident that the long-term memory is the one helping to obtain better results in stage 3. A more correct version of the experiment would be to ablate the tokens retrieved from the memory (i.e., change to padding, zero vectors, or noise). This would diminish any effect of the additional training in Stage 3.

**Methods And Evaluation Criteria:**

The methods and the evaluation criteria make sense for the presented application.
Note that no metric is described for the LongBench results (Table 2). Moreover, given those results, it is unclear what the authors want to convey with that experiment.
Also, it is worth mentioning that there are existing datasets that aim to test the quality of memory augmented models like M+ (see [1]).


[1] “Assessing Episodic Memory in LLMs with Sequence Order Recall Tasks” by Pink et al., 2024.

**Other Comments Or Suggestions:**

* Line 156: “tokens with the largest ages” -> “oldest tokens”
* Figure 3 shows “Llama-3.2-3B…” while the text mentions “Llama-3.1-3B”.

**Other Strengths And Weaknesses:**

* The work is clearly written and nicely presented
* The experiments evaluate a vast amount of details about the model

**Questions For Authors:**

* What is the metric used for LongBench?
* What is the performance of M+ on the task of [1]?

[1] “Assessing Episodic Memory in LLMs with Sequence Order Recall Tasks” by Pink et al., 2024.

**Relation To Broader Scientific Literature:**

The contributions in this work are relevant to this community and valuable. We should note that similar solutions have been proposed and have been shown to work similarly. This work doesn’t compare results beyond its predecessor (MemoryLLM).

**Theoretical Claims:**

N/A

---

> ### Author Rebuttal · Authors · 2025-04-01
>
> We sincerely thank reviewer f5Uz for their recognition of the value of our work. We address the reviewer's concerns below:
>
> **Relation To Broader Scientific Literature:**
>
> To the best of our knowledge, following MemoryLLM, the most recent works on parametric memory include Titans [3] and Memory at Scale [4], which scale models to 760M and 1.3B parameters respectively. However, these efforts remain exploratory and have not yet been evaluated on real-world tasks such as long-context question answering, which is the focus of our paper. As a result, there are currently few parametric memory approaches that offer a meaningful comparison to our work.
>
> As for the questions:
>
> **[Q1] Evaluation Metric in LongBench:**
>
> Thank you for pointing this out. Following LongBench [1], we use the QA-F1 Score as the evaluation metric for all six benchmarks included in our paper. We will add this clarification in the revised version of our manuscript.
>
> **[Q2] Related Work and Comparison with [2]:**
>
> We appreciate you highlighting [2]—Assessing Episodic Memory in LLMs with Sequence Order Recall Tasks. It is indeed a relevant and important contribution. We note that it is conceptually similar to our newly proposed Longbook-Event-QA task (Section 4.1.1). While [2] evaluates models by asking them to sort two segments after reading an entire book, Longbook-Event-QA requires selecting the next event given a sequence of events. Both tasks emphasize the model’s comprehension of the narrative and its ability to reason about event order. Given their shared focus, we anticipate consistent results between Longbook-Event-QA and [2], and we will consider incorporating [2] as an extended benchmark in future work.
>
> References:
>
> [1] LongBench: A Bilingual, Multitask Benchmark for Long Context Understanding.
> [2] Assessing Episodic Memory in LLMs with Sequence Order Recall Tasks.
> [3] Titans: Learning to Memorize at Test Time.
> [4] Memory Layers at Scale.

---

### Official Review · Reviewer_WXyo · 2025-03-16

**Overall Recommendation:** 3

**Summary:**

**Main Findings:**

Equipping large language models (LLMs) with latent-space memory has gained significant interest, as it extends the effective context window of existing models. However, preserving and retrieving information from distant past contexts remains challenging. To address this, this paper proposes M+, an enhancement of the existing MemoryLLM model, introducing a latent-space retrieval mechanism designed specifically for long-term memory retention.

**Main Algorithmic/Conceptual Ideas:**

M+ incorporates a co-trained retriever within the MemoryLLM framework, enabling dynamic retrieval of relevant latent space information during text generation. This allows the model to effectively leverage long-term contextual memories.

**Main Results:**

Empirical evaluations on several long-text benchmarks (including Longbook, SQuAD, and LongBench) demonstrate that M+ outperforms MemoryLLM.

**Claims And Evidence:**

Yes, the claims are clear and convincing.

**Essential References Not Discussed:**

[1] is one of the earliest works to discuss the design of extra long-term memory. [2] [3] are closely related and up-to-date works that design memory modules for long-context ability in LLMs.

[1] Hybrid computing using a neural network with dynamic external memory

[2] Titans: Learning to Memorize at Test Time

[3] Scaling Transformer to 1M tokens and beyond with RMT

**Experimental Designs Or Analyses:**

Yes, I have checked the experimental designs.

**Methods And Evaluation Criteria:**

Yes, the proposed methods make sense for the target problems.

**Other Comments Or Suggestions:**

No additional comments.

**Other Strengths And Weaknesses:**

**Strengths:**

The proposed method clearly demonstrates improvements over the original MemoryLLM by introducing an effective retrieval mechanism (M+) for handling extremely long-context content.

The paper provides detailed descriptions of algorithms and experimental setups, facilitating straightforward reimplementation by the research community.

**Weakness**:

1. The novelty of this paper is incremental. The proposed long-term memory mechanism closely resembles existing retrieval-based designs for long-context processing, such as SnapKV, into MemoryLLM. The primary difference is utilizing KV or latent states for long-term memory, which has not been clearly demonstrated as crucial for improving performance.

2. The paper lacks a rigorous analysis or ablation study comparing KV and latent states as long-term memory.

**Questions For Authors:**

1. Could you elaborate further on any additional conceptual or algorithmic insights beyond introducing a retrieval mechanism into MemoryLLM? Specifically, what distinct advantages or innovations does your method offer compared to existing retrieval-based approaches (e.g., SnapKV)?

2. It would be beneficial to explicitly compare the effectiveness of leveraging KV cache versus latent states as long-term memory. Providing experimental results or ablation studies on this comparison could clarify the significance and necessity of your design choice.

**Relation To Broader Scientific Literature:**

Introducing a retrieval mechanism into MemoryLLM can effectively address the challenges associated with extremely long-context content.

**Theoretical Claims:**

Not applicable. No theoretical claims.

---

> ### Author Rebuttal · Authors · 2025-04-01
>
> **Essential References Not Discussed:**
>
> Thank you for highlighting these important works. We will incorporate [1], [2], and [3] into our related work section. Specifically, [1] aligns with the core motivation behind incorporating memory into language models. Both [2] and [3] explore architectural modifications to enable memory mechanisms in transformers. However, these approaches remain exploratory, as they have not been scaled beyond small models or applied to real-world tasks such as long-context question answering. In contrast, M+ is implemented at the 8B parameter scale and is designed to be scalable with additional GPU resources. We will update our related work section accordingly and include a discussion of these points in the paper.
>
> **Weaknesses**:
>
> **[W1] Similarity to Attention-Based Retrieval Methods:** We acknowledge that our method shares some similarities with prior approaches that use attention to retrieve keys and values. However, there are critical differences that make our approach unique and practically advantageous:
>
> **(1) Efficiency:** Methods such as SnapKV maintain and retrieve key-value pairs per head, which becomes extremely costly when scaled. In our setting—with 32 layers and 32 attention heads per layer—this requires 1024 retrievals per query, resulting in significant latency (as noted in line 59 of our paper). In contrast, M+ uses a co-trained retriever to retrieve memory tokens, which are compressed hidden states. This results in only 32 total retrievals—one per layer—dramatically reducing both computational cost and latency.
>
> **(2) Performance:** In Figure 6, the curve labeled MemoryLLM-8B-Attn follows the SnapKV-style approach of retrieving key-value pairs using attention per head. As shown in the figure, it performs substantially worse than M+, highlighting that our co-trained retriever not only improves efficiency but also yields better results in practice compared with attention-based retrievals.
>
> **(3) Design:** Note that our training setup includes both relevant and irrelevant documents (See details in Appendix D), making it well-suited for contrastive learning. This allows us to effectively train the retriever, which integrates naturally into our overall training framework.
>
> **[W2] Representation of Long-Term Memory (Hidden States vs. KV):**
>
> We appreciate the reviewer’s insightful comment regarding the form of long-term memory. We advocate for the use of hidden states over key-value (KV) caches based on two key considerations:
>
> **Compression Efficiency:** As detailed in the paper, we compress each 512-token chunk into 256 memory vectors per layer in a lossless manner. In contrast, KV-based methods often require downsampling—e.g., dropping half the keys and values—to control memory size, resulting in unavoidable information loss.
>
> **Retrieval Efficiency and Performance:** As described in [W1],  hidden states can be effectively retrieved using our co-trained retriever, requiring only 32 retrievals for each query. In contrast, a KV-cache approach would demand up to 1024 retrievals, significantly increasing computational cost. Furthermore, as shown in Figure 6, using hidden states yields better performance compared to using KV caches.
>
> We believe these benefits make hidden states a more efficient and effective choice for long-term memory representation in our system.
>
> [1] Hybrid computing using a neural network with dynamic external memory.
> [2] Titans: Learning to Memorize at Test Time.
> [3] Scaling Transformer to 1M tokens and beyond with RMT.

---

> > ### Comment · Reviewer_WXyo · 2025-04-04
> >
> > I acknowledge the explanation provided in the text. However, I still believe that an apple-to-apple numerical comparison is necessary to demonstrate the effectiveness clearly. Although the authors state that “Furthermore, as shown in Figure 6, using hidden states yields better performance compared to using KV caches,” the comparison in Figure 6 is between MemoryLLM-8B and M+, which does not make it clear to the audience which part corresponds to an apple-to-apple comparison between KV cache and latent states. More explanations are necessary.

---

> > > ### Author Response · Authors · 2025-04-05
> > >
> > > We appreciate the reviewer’s suggestion regarding an apple-to-apple comparison between key-value caches and latent states.
> > >
> > > **[TL, DR] We would like to highlight that we provide the comparison between key-value caches and M+ in Figure 3, which is the comparison between Llama-3.1-8B-SnapKV and M+ (For Longbook-QA, M+ vs Llama-3.1-8B-SnapKV is 0.1752 vs 0.1625, For Longbook-Event-QA, M+ vs Llama-3.1-8B-SnapKV is 0.2476 vs 0.2297).**
> > >
> > > Specifically, we offer more explanations below:
> > >
> > > Based on our understanding, the comparison the reviewer is requesting can be framed through the following four settings:
> > >
> > > - Setting 1: Save the hidden states of $x_1, \cdots, x_n$, and use the question $q$ to retrieve some of these hidden states.
> > > - Setting 2: Save the keys and values of $x_1, \cdots, x_n$, and retrieve relevant keys and values per head using $q$. (**key-value caches applet-to-apple**  comparison with **M+**)
> > > - Setting 3: Compress $x_1, \cdots, x_n$ into hidden states, and use a co-trained retriever to fetch relevant hidden states given $q$. (**M+**)
> > > - Setting 4: Compress $x_1, \cdots, x_n$ into hidden states, transform them into keys and values, and use $q$ to retrieve keys and values.
> > >
> > > We hope this breakdown aligns with the reviewer’s intent. If there are additional settings the reviewer would like us to consider, we would be happy to incorporate them.
> > >
> > > Below, we explain how each of these settings corresponds to the methods evaluated in our paper:
> > > - Setting 1 is primarily used in early work such as KNN-LM [1], and has since been superseded by methods like H2O [2] and SnapKV [3]. While we did not include Setting 1 in our comparisons, we note that it is no longer used in recent literature, and thus we followed this trend.
> > > - Setting 2 corresponds to SnapKV, i.e. **Llama-3.1-8B-SnapKV** in Figure 3, where we implemented SnapKV on Llama-3.1-8B, and we present a direct comparison between SnapKV and our method (Llama-3.1-8B-SnapKV vs M+).
> > > - Setting 3 corresponds to M+, which is compared with both key-value cache baselines and MemoryLLM.
> > > - Setting 4 corresponds to our model MemoryLLM-8B-Attn, as shown in Figure 6.
> > >
> > > To summarize, our paper includes:
> > > - A comparison between Setting 2 and Setting 3 (key-value caches vs. M+), and
> > > - A comparison between Setting 3 and Setting 4 (M+ vs. MemoryLLM).
> > >
> > > Therefore, we believe the paper covers all meaningful and contemporary comparisons between key-value caches and latent state representations. We acknowledge the reviewer’s request and hope this clarification addresses the concern.
> > >
> > > Finally, we would like to clarify that the “hidden states” saved in our system are not equivalent to those in Setting 1. In our case, the hidden states are outputs of an encoder that is trained jointly with the rest of the system. As a result, these are compressed representations and contain less redundant information compared with traditional decoder-side hidden states in a language model.
> > >
> > > We thank the reviewer again for the thoughtful feedback and are happy to provide additional clarifications if needed. Meanwhile, if the reviewer finds our response adequately addresses the concern, **we would be sincerely grateful if they would consider reassessing the evaluation and potentially raising their score.**
> > >
> > > [1] Generalization through Memorization: Nearest Neighbor Language Models.
> > > [2] H2O: Heavy-Hitter Oracle for Efficient Generative Inference of Large Language Models.
> > > [3] SnapKV: LLM Knows What You are Looking for Before Generation.

---

### Decision · Program_Chairs · 2025-05-01

**Decision:**

Accept (poster)

**Comment:**

This paper builds on MemoryLLM by introducing M+, a memory‑augmented transformer that integrates (1) a compressed latent‑space long‑term memory pool and (2) a co‑trained retriever to fetch relevant memory tokens during generation. By storing and retrieving 256 compressed vectors per layer rather than per‑head key‑value pairs, M+ reduces retrieval overhead (32 vs. 1,024 retrievals per query) and extends effective context retention from ~20 K to over 160 K tokens. The authors evaluate on long‑context QA (LongBook‑QA, LongBook‑Event‑QA), standard QA (SQuAD, NaturalQA), and multi‑domain long‑context benchmarks (LongBench), reporting consistent gains over LLaMA‑3.1, MemoryLLM, SnapKV, and RAG baselines. Ablations confirm the benefits of the retrieval mechanism and hidden‑state representation.

Reviewers generally agree that M+ delivers solid, practically relevant improvements in long‑context retention. Concerns about incremental novelty are outweighed by the paper’s scale, efficiency, and thorough empirical validation. The authors’ rebuttal clarified comparisons to KV‑cache baselines (via additional experiments in Figures 3, 5, 6), detailed memory/latency scaling, and provided short‑document performance to contextualize gains. Requests for deeper interpretability and theoretical analysis fall outside this paper’s primarily empirical scope but can be addressed in follow‑up work.